# Safe Explicable Planning

**Primary Keywords:** *Human-aware Planning and Scheduling*

## Abstract

Human expectations stem from their knowledge about the others and the world. Where human-AI interaction is concerned, such knowledge may be inconsistent with the ground truth, resulting in the AI agent not meeting its expectations and degraded team performance. Explicable planning was previously introduced as a novel planning approach to reconciling human expectations and the agent's optimal behavior for more interpretable decision-making. One critical issue that remains unaddressed is safety in explicable planning since it can lead to explicable behaviors that are unsafe. We propose *Safe Explicable Planning (SEP)* to extend the prior work to support the specification of a safety bound. The objective of SEP is to search for behaviors that are close to the human's expectations while satisfying the bound on the agent's return, the safety criterion chosen in this work. We show that the problem generalizes the consideration of multiple objectives to multiple models and our formulation introduces a Pareto set. Under such a formulation, we propose a novel exact method that returns the Pareto set of safe explicable policies, a more efficient greedy method that returns one of the Pareto optimal policies, and approximate solutions for them based on the aggregation of states to further scalability. Formal proofs are provided to validate the desired theoretical properties of the exact and greedy methods. We evaluate our methods both in simulation and with physical robot experiments. Results confirm the validity and efficacy of our methods for safe explicable planning.

## INTRODUCTION

The capabilities of AI agents have advanced rapidly in recent years, to the extent that they are no longer confined to a space of their own but deployed in environments surrounded by humans. Examples of such agents include Starship's food delivery robots, Amazon's Astro - a household robot, Bear Robotics' hospitality robots, Waymo's autonomous driving cars, and many others. As these agents further develop, they are expected to integrate into our daily lives and become our partners. In such situations, it is required for them to behave as in human-human collaboration where one of the critical aspects is for behaviors to align with others' expectations.

Existing work that considers human expectation in decision-making is referred to as explicable planning (Zhang et al. 2017; Kulkarni et al. 2016; Hanni and Zhang 2021). It is assumed that the humans generate their expectations

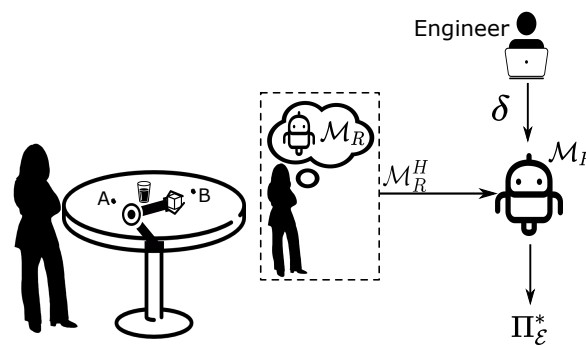

(a) Motivating example     (b) Problem setting

Figure 1: The agent uses the ground-truth model $\mathcal{M}_R$, (an estimation of) the human's understanding of it, $\mathcal{M}_R^H$, and a bound $\delta$, to generate safe explicable policies $\Pi_{\mathcal{E}}^*$.

of an agent's behavior based on their understanding of the agent and the world ($\mathcal{M}_R^H$), which may differ from the ground truth (the agent's model or $\mathcal{M}_R$) (see Fig. 1(b)). In the original formulation, the objective is to search for a plan that maximizes its similarity to the human's expected plan, as measured by a new explicability metric, while simultaneously minimizing a plan cost metric via a linearly weighted sum of the two metrics. To handle stochastic domains, (Gong and Zhang 2022) optimizes a similar objective under Markov Decision Process (MDP) in a learning setting. An important limitation of these existing approaches is that they do not bound the suboptimality of the solution under the ground-truth model (i.e., $\mathcal{M}_R$). This is due to the fact that the trade-off between cost and explicability metrics (at different scales) relies on a hyper-parameter, referred to as the reconciliation factor by (Zhang et al. 2017). Consequently, generating an explicable behavior may result in over-compromising the cost in the ground-truth model, resulting in unsafe behaviors.[1] Intuitively, a solution to such a problem would be to extend explicable planning by incorporating a bound on the cost in the ground-truth model.

Let us further illustrate the need for safe explicable planning (SEP) via a motivating scenario. Consider a human user

---

[1] An underlying assumption here is that safety is negatively correlated with the cost metric. Other forms of safety criteria, such as behavior deviation, will be considered in future work.

working beside a robot manipulator. The robot is required to hand over a box to the human by placing it at either location '*A*' or '*B*', as depicted in Fig. 1(a). Location '*A*' is closer to the human but can lead to the robot tipping over a water cup with a small probability. When the cup is empty, the cost from tipping over the cup can be safely ignored. In such a case, the desired behavior would be for the robot to place the box at '*A*' to better align with the human's expectation due to its proximity. However, when the cup is nonempty, the desired behavior is to place the box at '*B*' to avoid tipping over the cup since it can introduce a hazard of electric shock and a significant cost under the robot's model ($\mathcal{M}_R^H$). The difference, however, is so subtle that it may not be obvious from the human's perspective (based on the human's model $\mathcal{M}_R^H$). The result could be the robot performing "explicably" to conform with the human's expectation under both cases using explicable planning, leading to a safety risk. In SEP, due to the bound on the cost in the robot's model, the robot would never choose the unsafe behavior.

We make the following assumptions in developing our approach to SEP to focus on the planning challenges. First, we assume the agent has access to both $\mathcal{M}_R$ and $\mathcal{M}_R^H$ as in many prior works on explicable planning (Kulkarni et al. 2016; Hanni and Zhang 2021) and explainable decision-making (Chakraborti et al. 2019), in general. The human's model $\mathcal{M}_R^H$ may be obtained via learning from human feedback (e.g., (Christiano et al. 2017; Ibarz et al. 2018; Holmes et al. 2004; Juba and Stern 2022)). Second, we assume the human is a rational observer: expectations of the agent are generated by computing the optimal behavior under the human's model ($\mathcal{M}_R^H$). It also reduces the maximization of the explicability of a behavior to the maximization of its return under $\mathcal{M}_R^H$ (when modeled as an MDP). The assumption of human rationality is a common simplification in cognitive science (Baker, Saxe, and Tenenbaum 2011) and AI.

We formulate SEP under MDPs. First, we define the objective as maximizing the expected return in $\mathcal{M}_R^H$ subject to a constraint on $\mathcal{M}_R$ specified by the bound $\delta$. This problem formulation generalizes the consideration of multiple objectives (Marler and Arora 2004) to include multiple domain models. The solution is a Pareto set of policies for which exact solvers are generally intractable. We first develop an action pruning technique that significantly reduces the policy space. Then, we introduce a novel tree search method that efficiently searches through the remaining policies to identify the Pareto set. We formally prove that such a search method is sound and complete. In case any policy from the Pareto set is sufficient, we further introduce a greedy search method. Finally, we create approximate solutions for both search methods via state aggregation to scale them to complex domains. For evaluation, we investigate our methods on several domains in simulation and with physical robot experiments to demonstrate their efficacy for SEP. In addition, we analyze the benefits of our pruning techniques via ablation studies to validate their effectiveness.

## RELATED WORK

There has been a growing interest in explainable decision-making to develop AI agents whose behaviors are explainable to humans (Chakraborti et al. 2019; Chakraborti, Sreedharan, and Kambhampati 2020; Fox, Long, and Magazzeni 2017). We may broadly classify methods in this area into two classes: those that generate behaviors that are more interpretable (implicitly explainable) and those that communicate to explain behaviors (explicitly explainable). Our work belongs to the former. Researchers have approached implicit explainable decision-making from various but related perspectives; generating behaviors that are considered legible (Dragan and Srinivasa 2013), predictable (Dragan and Srinivasa 2013), transparent (MacNally et al. 2018), explicable (Zhang et al. 2017), etc. A review of their relationships is provided by (Chakraborti et al. 2019). Our work extends explicable planning by addressing an important gap in applying such methods to the real world.

Our problem formulation of safe explicable planning (SEP) has close connections to the constrained-criterion-based formulation in safe reinforcement learning (RL) (García and Fernández 2015) that inherently models the problem as a Constrained MDP (CMDP) (Altman 2021). Generally, safety is encoded by constraining the expected cost under some cost function given in addition to the agent's reward function. In our work, we encode safety by directly constraining the expected return under the agent's reward function. We assume that safety is correlated to the expected return in the agent's model under the intuition that unsafe behaviors would result in low returns. Our formulation can easily consider a CMDP problem by aligning the two models and imposing the safety constraint on a separate cost function (i.e., substituting the robot's reward function in the constraint with the cost function).

A unique challenge in formulating SEP under CMDP is the presence of two separate MDP models. More specifically, apart from the models having two different reward functions, we consider a slightly general setting that allows the two models to have different domain dynamics and discount factors as well. Such a general setting makes existing solution methods for CMDP inapplicable. For example, consider the linear programming (LP) based solution for CMDP (Altman 1994). The LP objective is defined by an occupation measure, for different state action pairs, which depends on the transition model and the discount factor. When the models are different, applying the LP solution to SEP would result in a different occupation measure being used in the objective from that used in the constraints and it is not straightforward to solve for these two sets of variables. Similar arguments can be made about the other solution methods.

The objective considered in SEP is also related to Multi-Objective Markov Decision Processes (MOMDP) (Wakuta and Togawa 1998), in the sense that SEP considers both the expected return of the agent's reward and the human's belief of the agent's reward. Since MOMDPs consider multiple objectives under the same MDP model, the solutions proposed (refer to the review paper by (Roijers et al. 2013)) aim to optimize a vector of expected returns from multiple objectives to obtain a Pareto set of solutions or obtain a single solution by considering a linear scalarization of objectives. When the models are different, applying MOMDP techniques would result in multiple vectors (one for each model) of expected

returns from multiple objectives. Optimizing these vectors simultaneously is substantially more challenging than optimizing a single vector in MOMDPs since they are computed from different models. While general studies in MOMDP do not consider constraints, a specific study by (Wray, Zilberstein, and Mouaddib 2015) considers constraint specifications for multiple objectives with a lexicographic ordering which has close connections to our work and has inspired the action pruning technique described in this paper. However, in addition to the limitation above, the solution does not guarantee the optimality of the policy found.

There exists prior work that considers multiple MDPs (Singh and Cohn 1997; Russell and Zimdars 2003; Buchholz and Scheftelowitsch 2019) which primarily focus on finding a policy that maximizes the combined or weighted expected return from all reward functions, essentially reducing it to a single objective optimization problem. Even though these methods may appear comparable to ours, they can result in a policy that violates the safety bound or has low quality in the human's model. Such an issue is due to the fact that these methods do not explicitly consider safety bounds, which we address in our work.

## PROBLEM FORMULATION

In safe explicable planning, there are two models at play: $\mathcal{M}_R$ and $\mathcal{M}_R^H$. We formulate these models as discrete Markov Decision Processes (MDPs). An MDP is represented by a tuple $\mathcal{M} = \langle \mathcal{S}, \mathcal{A}, \mathcal{T}, \mathcal{R}, \gamma \rangle$ where $\mathcal{S}$ is a set of states, $\mathcal{A}$ is a set of actions, $\mathcal{T}(s'|s, a)$ is a transition function, $\mathcal{R}$ is a reward function, and $\gamma$ is a discount factor. We assume $\mathcal{M}_R$ and $\mathcal{M}_R^H$ share the same state and action spaces but have different transition functions, reward functions, and discount factors. This is reasonable when the human and AI agent cohabit the workspace and share certain understanding of the environment. Relaxing such an assumption incurs separate technical challenges (e.g., hierarchical models) that will be deferred to future work. Hence, the ground-truth or agent's model $\mathcal{M}_R$ is represented by the tuple $\mathcal{M}_R = \langle \mathcal{S}, \mathcal{A}, \mathcal{T}_R, \mathcal{R}_R, \gamma_R \rangle$ where $\mathcal{T}_R$ is the true transition function or domain dynamics, $\mathcal{R}_R$ is the engineered reward function, and $\gamma_R$ is the engineered discount factor. The human's understanding or belief about the agent's model $\mathcal{M}_R^H$ is represented by the tuple $\mathcal{M}_R^H = \langle \mathcal{S}, \mathcal{A}, \mathcal{T}_R^H, \mathcal{R}_R^H, \gamma_R^H \rangle$ where $\mathcal{T}_R^H$ is the human's belief about $\mathcal{T}_R$, $\mathcal{R}_R^H$ is the belief about $\mathcal{R}_R$, and $\gamma_R^H$ is the belief about $\gamma_R$.

We work with the set of all stationary deterministic policies $\Pi$, given by $\forall \pi \in \Pi, \pi : \mathcal{S} \mapsto \mathcal{A}$. An optimal agent's policy maximizes the (expected) return in the agent's model and is given by $\pi^* = \arg\max_\pi \mathbb{E}_{\mathcal{T}_R}^\pi [\sum_{t=0}^\infty \gamma_R^t r_R(t)]$. We define a safe behavior as any behavior with a return within a bound of the optimal agent's return. Similar criteria have been used in safe RL (García and Fernández 2015; Moldovan and Abbeel 2012). More formally, a policy $\pi$ is considered safe if its return satisfies the following condition:

$$\mathbb{E}_{\mathcal{T}_R}^\pi \left[ \sum_{t=0}^\infty \gamma_R^t r_R(t) \right] \geq \delta \mathbb{E}_{\mathcal{T}_R}^{\pi^*} \left[ \sum_{t=0}^\infty \gamma_R^t r_R(t) \right], \quad (1)$$

where $\delta \in (0, 1]$ is the designer-specified safety bound.

Since execution may start from any state, we require such a condition to hold true under *any state*. It also implies that the condition would hold from any step during execution. These are desirable features of safety critical systems.

In prior work on explicable planning, the objective has been to maximize a weighted sum of the return in the agent's model and an explicability metric. For example, such a metric has been defined via plan distances (Kulkarni et al. 2016) in deterministic domains and KL divergence between trajectory distributions (Gong and Zhang 2022) in stochastic domains. In our work, we define the explicability metric simply as the return from the human's model $\mathcal{M}_R^H$. Given that the human uses $\mathcal{M}_R^H$ to generate expectations, this assumes rational human observer: the higher the return in the human's model, the more expected the policy is.

**Definition 1.** Safe Explicable Planning (SEP), given by $\mathcal{P}_\mathcal{E} = \langle \mathcal{M}_R, \mathcal{M}_R^H, \delta \rangle$, is the problem to search for a policy that maximizes the return in $\mathcal{M}_R^H$ subject to a constraint on the return in $\mathcal{M}_R$ under *any state*, or more formally:

$$\pi_\mathcal{E}^* = \arg\max_\pi \mathbb{E}_{\mathcal{T}_R^H}^\pi \left[ \sum_{t=0}^\infty \gamma_R^{H\,t} r_R^H(t) \right] \text{ subject to}$$

$$\mathbb{E}_{\mathcal{T}_R}^\pi \left[ \sum_{t=0}^\infty \gamma_R^t r_R(t) \right] \geq \delta \mathbb{E}_{\mathcal{T}_R}^{\pi^*} \left[ \sum_{t=0}^\infty \gamma_R^t r_R(t) \right]. \quad (2)$$

Requiring the constraint above to hold under any state introduces a Pareto set of optimal policies where no policies in this set are strictly dominated by any policy. Briefly, a policy $\pi_1$ strictly dominates another policy $\pi_2$ if its state values are no smaller in any state, and larger in at least one state. More formally, denote such a relationship as $\pi_1 \succ \pi_2$, which holds if $\forall s \in \mathcal{S}\ [V_{\mathcal{M}_R^H}^{\pi_1}(s) \geq V_{\mathcal{M}_R^H}^{\pi_2}(s)] \wedge \exists s' \in \mathcal{S}\ [V_{\mathcal{M}_R^H}^{\pi_1}(s') > V_{\mathcal{M}_R^H}^{\pi_2}(s')]$. The Pareto set $\Pi_\mathcal{E}^*$ is then given by:

$$\Pi_\mathcal{E}^* = \{\pi_\mathcal{E}^* \in \Pi_\delta \mid \neg \exists \pi \in \Pi_\delta[\pi \succ \pi_\mathcal{E}^*]\}, \quad (3)$$

where $\Pi_\delta = \{\pi \in \Pi \mid \forall s \in \mathcal{S}\ [V_{\mathcal{M}_R}^\pi(s) \geq \delta V_{\mathcal{M}_R}^{\pi^*}(s)]\}$ is the set of policies that satisfy the safety bound.

## SAFE EXPLICABLE PLANNING

In this section, we motivate and discuss our solution methods for SEP. Given the large policy space to search for, we first discuss a technique to cut the policy space. Since any policy in $\Pi_\delta$ may be in the Pareto set, we are required to expand all policies in $\Pi_\delta$. We propose an exact method that almost expands only those policies in $\Pi_\delta$ to determine the Pareto set $\Pi_\mathcal{E}^*$. A greedy method that only expands a subset of policies in $\Pi_\delta$ and returns a single policy in $\Pi_\mathcal{E}^*$ is then discussed. Finally, we propose approximate solutions via state aggregation using handcrafted features to condition similar states to choose the same actions to further scalability. Complete proofs are provided in the supplemental materials.

### Policy Space Reduction via Action Pruning

Even though the set of $\Pi_\delta$ cannot be obtained directly from the entire policy space $\Pi$, we aim to cut the policy space based on the safety constraint to produce a subset of policies

in $\Pi$, referred to as $\widetilde{\Pi}$. The challenge here is to ensure that $\widetilde{\Pi} \supseteq \Pi_\delta$ (see Fig. 2(a)).

We achieve this by pruning sub-optimal actions for every state that are guaranteed to violate the constraint. More specifically, let $\mathcal{A}(s)$ be the set of all actions that are available in state $s$. The set of actions after pruning is given by:

$$\widetilde{\mathcal{A}}(s) = \{a \in \mathcal{A}(s) | Q_{\mathcal{M}_R}^{\pi^*}(s, a) \geq \delta \max_{a' \in \mathcal{A}(s)} Q_{\mathcal{M}_R}^{\pi^*}(s, a')\}. \tag{4}$$

The policy space after action pruning for all states is $\widetilde{\Pi}$. Our action pruning technique is inspired by (Wray, Zilberstein, and Mouaddib 2015; Pineda, Wray, and Zilberstein 2015). To provide a worst-case guarantee (where all states choose actions as far as possible from the optimal after $a$) under $\mathcal{M}_R$, the authors used $1 - (1 - \gamma)(1 - \delta)$ instead of $\delta$ in Eqn. (4), resulting in a different set of policies, referred to as $\Pi_\eta$. Their pruning condition is more stringent than ours and may result in pruning state-actions that belong to policies satisfying the constraint in Eqn (2). Consequently, the guarantee that $\Pi_\eta \supseteq \Pi_\delta$ is lost there (see Fig. 2(a)).

**Lemma 1.** The set of policies after action pruning based on Eqn. (4) is a superset of the set of policies that satisfy the constraint in Eqn. (2), i.e., $\widetilde{\Pi} \supseteq \Pi_\delta$.

**Proof Sketch:** To prove this result, we show that an action pruned by any state per Eqn. (4) is guaranteed to introduce policies that do not satisfy the constraint in Eqn. (2) under at least one state. We show that the expected return of choosing a pruned action in that state and then following the optimal policy thereafter does not satisfy the constraint. Hence, any policy that chooses the pruned action for that state cannot satisfy the constraint either.

## Policy Descent Tree Search (PDT)

To determine $\Pi_\mathcal{E}^*$, intuitively, we can evaluate every policy in $\widetilde{\Pi}$. However, this would be impractical and proves to be unnecessary. A better idea is to enable further pruning in $\widetilde{\Pi}$ by expanding policies in certain order that facilitates pruning. We consider two options here. First, we can start from the optimal policy in the human's model and apply policy ascent in the agent's model to improve it until the bound is satisfied. Alternatively, we can start from the agent's optimal policy and apply policy descent in the agent's model and identify better policies in the human's model until the bound is violated. Since the first search strategy can lead to missed policies in $\Pi_\mathcal{E}^*$, we choose the latter option in our work.

In tree search, we start from an optimal policy in $\mathcal{M}_R$, denoted by $\pi^*$, as the root node. The benefit of doing so is that, first, we already know that $\pi^*$ satisfies the bound under the agent's model since it is the optimal policy under $\mathcal{M}_R$. Second, we can leverage the known state values $V_{\mathcal{M}_R}^{\pi^*}$ to expand policies that have lower state values than that of the parent node recursively. Since this is the opposite of policy improvement, we refer to it as policy descent. More formally, all descendants of a policy $\pi$ under single-action policy updates in PDT can be obtained by replacing $\pi(s)$ under *any*

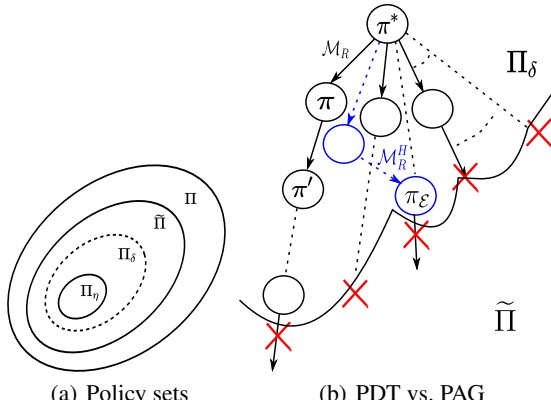

(a) Policy sets      (b) PDT vs. PAG

Figure 2: (a) Relationship between the different policy sets. (b) PDT vs. PAG on action-pruned space $\widetilde{\Pi}$. The black nodes are expanded by PDT in descending order of state values under $\mathcal{M}_R$. The blue nodes are expanded by PAG in ascending order under $\mathcal{M}_R^H$. Solid lines represent single-action policy updates and dashed links represent multi-action updates.

state $s$ with an action $a$ that satisfies:

$$\sum_{s'} \mathcal{T}_R(s, a, s') V_{\mathcal{M}_R}^\pi(s') \leq \sum_{s'} \mathcal{T}_R(s, \pi(s), s') V_{\mathcal{M}_R}^\pi(s'). \tag{5}$$

Once a branch reaches a policy whose state values no longer satisfy the bound under $\mathcal{M}_R$ (any state suffices), it is pruned as illustrated in Fig. 2(b). The search continues until all branches are pruned and the set of non-dominated policies under $\mathcal{M}_R^H$ are maintained. The algorithm is presented in Alg. 1, which we refer to as PDT+ (action pruning). Next, we formally show that such a process returns $\Pi_\mathcal{E} = \Pi_\mathcal{E}^*$.

**Lemma 2.** Let $\pi$ and $\pi'$ be two deterministic policies that differ by only a single action in some state i.e., $\exists s_i \in \mathcal{S}$ $[\pi'(s_i) \neq \pi(s_i)] \wedge \forall s_j \in \mathcal{S} \setminus \{s_i\} [\pi'(s_j) = \pi(s_j)]$ and satisfy $Q_{\mathcal{M}_R}^\pi(s_i, \pi'(s_i)) \leq V_{\mathcal{M}_R}^\pi(s_i)$. Then, policy $\pi'$ is a descendant of $\pi$ in PDT, i.e., policy $\pi'$ is no better than $\pi$, or more formally, $\forall s \in \mathcal{S} [V_{\mathcal{M}_R}^{\pi'}(s) \leq V_{\mathcal{M}_R}^\pi(s)]$.

**Proof Sketch:** This is an extension of the policy improvement theorem (Sutton and Barto 2018) but in the opposite direction (hence referred to as a policy descent step). We first introduce a temporary non-stationary policy $\pi'_1$ that chooses an action as per $\pi'$ under the initial state and follows $\pi$ thereafter. We can show that the return of $\pi'_1$ is no better than that of $\pi$. We can repeat such a pattern to update $\pi'_1$ for the next state and so on, resulting in $\pi'$ at the end.

Similarly, we can show that a special case of the policy improvement theorem holds when a single action is updated (referred to as a policy ascent step).

**Theorem 1.** PDT+ returns all Pareto optimal policies in $\Pi_\mathcal{E}^*$.

**Proof Sketch:** To prove this, we show that there exists a policy descent path from any optimal policy (denoted by $\pi^*$) in $\mathcal{M}_R$ (i.e., the root node in PDT) to any Pareto optimal policy (denoted by $\pi_\mathcal{E}^*$) in $\Pi_\mathcal{E}^*$ by induction. When $\pi_\mathcal{E}^*$ differs from $\pi^*$ in only 1 action, $\pi_\mathcal{E}^*$ must be one of the direct descendants of $\pi^*$ in PDT as $\pi^*$ is optimal in $\mathcal{M}_R$ based on the search

## Algorithm 1: PDT+

**Input**: $\mathcal{M}_R, \mathcal{M}_R^H, \delta$
$V_{\mathcal{M}_R}^* \leftarrow \texttt{ValueIteration}(\mathcal{M}_R)$; retrieve $\pi^*$
Compute $\widetilde{\mathcal{A}}(s), \forall s \in S$;
Initialize $\Pi_{\mathcal{E}} \leftarrow \emptyset$; $\texttt{fringe.push}(\pi^*)$;
**while** $\texttt{fringe} \neq \emptyset$ **do**
    $\pi \leftarrow \texttt{fringe.pop}()$;
    **for** $a$ *in* $\widetilde{\mathcal{A}}(s), s \in S$ **do**
        **if** *Eqn.* (5) *is satisfied* **then**
            $\pi' \leftarrow \texttt{Modify}(\pi, \pi(s) = a)$;
            **if** $\forall s \in S\,[V_{\mathcal{M}_R}^{\pi'}(s) \geq \delta V_{\mathcal{M}_R}^{\pi^*}(s)]$ **then**
                $\texttt{fringe.push}(\pi')$;
                **if** $\texttt{nonDominated}(\pi', \Pi_{\mathcal{E}}, \mathcal{M}_R^H)$ **then**
                    $\Pi_{\mathcal{E}}.\texttt{update}(\pi')$;

**return** $\Pi_{\mathcal{E}}$

---

process. Hence, $\pi_{\mathcal{E}}^*$ will be expanded by PDT. Assume any policy that differs from $\pi^*$ in $k$ actions are expanded. When $\pi_{\mathcal{E}}^*$ differs from $\pi^*$ in $k+1$ actions, we show that there must exist a policy $\pi$ that differs from $\pi^*$ in $k$ out of the $k+1$ actions (hence differing from $\pi_{\mathcal{E}}^*$ in 1 action) and (by Lem. 2) is no worse than $\pi_{\mathcal{E}}^*$ under $\mathcal{M}_R$ via proof by contradiction. Consequently, $\pi_{\mathcal{E}}^*$ must be a descendant of $\pi$ in PDT. Since $\pi$ is expanded under our inductive assumption, $\pi_{\mathcal{E}}^*$ will be expanded. Then by Lem. 1, the conclusion holds.

### Policy Ascent Greedy Search (PAG)

In certain situations, it may be unnecessary to compute $\Pi_{\mathcal{E}}^*$: any policy in the set would suffice. To this end, we introduce a greedy method that only searches through a subset of $\Pi_{\delta}$, making it more computationally efficient than PDT.

Similar to PDT, we start with $\pi^*$ at the root node. However, unlike in PDT where we expand policies that have a lower state values under $\mathcal{M}_R$ via single-action policy updates, we expand only a single policy that has higher values under $\mathcal{M}_R^H$ than its parent node via multi-action policy updates (see Fig. 2(b)). More formally, only one descendant of policy $\pi$ is expanded in PAG, which is obtained by replacing $\pi(s)$ under *each* state $s$ with an action $a$ that satisfies the following condition (similar to a policy improvement step):

$$\sum_{s'} \mathcal{T}_R^H(s, a, s') V_{\mathcal{M}_R^H}^{\pi}(s') \geq \sum_{s'} \mathcal{T}_R^H(s, \pi(s), s') V_{\mathcal{M}_R^H}^{\pi}(s'),$$
$$(6)$$

where each such state-action update is checked against the constraint in Eqn. (2) (under $\mathcal{M}_R$) incrementally and incorporated only if the constraint is not violated, resulting in a multi-action policy update for $V_{\mathcal{M}_R^H}^{\pi}$.

In PAG, we maintain a single candidate policy $\pi_{\mathcal{E}}$ as opposed to a set in PDT. The current policy $\pi_{\mathcal{E}}$ is updated to its descendant $\pi'$ if at least one of the state-action updates is incorporated. This process is repeated until $\pi_{\mathcal{E}}$ remains unchanged. The algorithm, referred to as PAG+ (action pruning), is presented in Alg. 2.

**Theorem 2.** PAG+ returns a policy in the Pareto set $\Pi_{\mathcal{E}}^*$.

**Proof Sketch:** The PAG search process stops when it can no longer improve or find a policy that is equivalent in values

## Algorithm 2: PAG+

**Input**: $\mathcal{M}_R, \mathcal{M}_R^H, \delta$
$V_{\mathcal{M}_R}^* \leftarrow \texttt{ValueIteration}(\mathcal{M}_R)$; retrieve $\pi^*$
Compute $\widetilde{\mathcal{A}}(s), \forall s \in S$;
Initialize $\pi_{\mathcal{E}} \leftarrow \pi^*$;
$\texttt{changed} \leftarrow true$;
**while** $\texttt{changed}$ **do**
    $V_{\mathcal{M}_R^H}^{\pi_{\mathcal{E}}} \leftarrow \texttt{PolicyEvaluation}(\pi_{\mathcal{E}}, \mathcal{M}_R^H)$
    $\texttt{changed} \leftarrow false$
    **for** $a$ *in* $\widetilde{\mathcal{A}}(s), s \in S$ **do**
        **if** *Eqn.* (6) *is satisfied* **then**
            $\pi' \leftarrow \texttt{Modify}(\pi_{\mathcal{E}}, \pi_{\mathcal{E}}(s) = a)$;
            **if** $\forall s \in S[V_{\mathcal{M}_R}^{\pi'}(s) \geq \delta V_{\mathcal{M}_R}^{\pi^*}(s)]$ **then**
                Update $\pi_{\mathcal{E}} \leftarrow \pi'$
                $\texttt{changed} \leftarrow true$

**return** $\pi_{\mathcal{E}}$

---

to $\pi_{\mathcal{E}}$ under $\mathcal{M}_R^H$ while satisfying the safety constraint. This translates to that there does not exist a state-action update that implements a policy ascent step under the constraint. However, if $\pi_{\mathcal{E}} \notin \Pi_{\mathcal{E}}^*$, there must exist another policy $\pi \in \Pi_{\mathcal{E}}^*$ that dominates $\pi_{\mathcal{E}}$, which contradicts with the fact that no policy ascent step exists. Then by Lem. 1, $\pi_{\mathcal{E}} \in \Pi_{\mathcal{E}}^*$.

### Approximate Solution via State Aggregation

In the worst case, the number of policies that PDT+ and PAG+ must search through is in the order of $|\widetilde{\Pi}|$, which is still exponential. Hence, it would be challenging to directly apply these methods to complex domains. Approximate solutions are needed. However, since the search is over the policy space, typical methods for function-approximating state value functions to search for optimal policies (Sỳkora 2008; Abel, Hershkowitz, and Littman 2016; Abel et al. 2018; Ferrer-Mestres et al. 2020) do not apply here.

We aim to design an approximate solution that can reduce the unique number of policies to be searched for. Inspired by function approximation, one idea is to reduce the state space by aggregating states that are alike in terms of action selection. The likeness of states can be measured by domain dependent features. We can condition states from the same clusters to choose the same actions under any policy with either model, effectively reducing the state space size and thus the number of policies. More formally, such a process introduces a mapping $\Phi : \mathcal{S}_K \mapsto \mathcal{S}$, a one-to-many mapping from clusters to states and $K$ is the number of clusters. Both PDT and PAG can work with the aggregated state space (i.e., clusters) by considering $\mathcal{S}_K$ as the new state space.

Under the assumption that the states in any aggregated state are "correlated" under any given policy for action selection, the same guarantees of optimality, completeness, and constraint satisfaction hold. Such a situation may occur, for example, when two states are topologically equivalent such that a reasonable policy should always choose the same action under these states. It would be interesting to study when such states are introduced, as well as the impact on the theoretical guarantees when such an assumption does not hold or hold only approximately. From such a per-

spective, our approximation method is analogous to function approximation under Q-learning.

# EVALUATION

We evaluate our methods under various domains in simulation and with physical robot experiments. The objective of our evaluations is threefold. First, we compare safe explicable behaviors with optimal behaviors to validate the effectiveness of our approach. Second, since the solution to SEP requires searching for the optimal policy in the feasible policy space in a brute-force manner to obtain the Pareto set, we analyze the efficiency of the proposed methods and compare with baselines (BF & BF+) that brute-force the policy search. Note that our comparisons are against the brute force methods because all prior studies discussed in the related work section are limited in terms of the consideration of multiple models or safety bounds (refer to related work). We also analyze the benefits of our approximate solutions with more complex domains and the action pruning technique in ablation studies for each proposed method. Third, we evaluate with physical robot experiments to show the applicability of our approach to real-world scenarios. By our naming convention, we append '+' to a method name to indicate the incorporation of our action pruning technique, such that the policy space is reduced to $\widetilde{\Pi}$; a method whose name is without '+' must deal with the original policy space, or $\Pi$ (see Fig. 2(a)). All evaluations were run on a MacBook Pro (16 GB, 3.1 GHz Dual-Core Intel Core i5). Details of the domain descriptions and implementation are in the supplemental materials.

*Bound Selection*: In our approach, the bound ($\delta$) is assumed to be specified by the designer based on experience. However, it can often be approximated based on the domain. For example, consider one of the cliff worlds in Fig. 5 (see below). The return of the optimal trajectory under the agent's model is 94 (i.e., moving along the edge of the cliff to the goal) and the return of the trajectory with the longest detour (i.e., staying as far away from the edge as possible) without falling off the cliff is 90, when ignoring any discount. Since unsafe behaviors should be captured by the agent's (ground-truth) model and result in much lower return than the detour, the safety bound can be set at $90/94 = 0.957$ and then adjusted. More analyses will be deferred to future work.

*Policy Selection*: To select from the Pareto set $\Pi_{\mathcal{E}}^*$, we can rely on user preferences. Alternatively, domain-specific scores may be introduced to assist with the selection. For example, higher scores may be given to policies that appear "simpler". Example scores are discussed when applicable.

## Simulations

**Domain Descriptions.** 1) *Cliff Worlds (CS & CL)*: The agent is required to navigate alongside the edge of a cliff to reach the goal (see Figs. 4 and 5). The ground-truth model ($\mathcal{M}_R$) is that the agent can travel alongside the edge without slipping off the cliff. The human's belief ($\mathcal{M}_R^H$) is that the agent may slip off from the edge with some probability, and the terrain closer to the cliff is more uneven and hence more difficult to traverse. For both worlds, the reward functions

| | $\delta$ | BF # | BF+ # | PDT # | PDT RT | PDT+ # | PDT+ RT | $\|\Pi_{\mathcal{E}}^*\|$ | PAG # | PAG RT | PAG+ # | PAG+ RT |
|---|---|---|---|---|---|---|---|---|---|---|---|---|
| CS | 1.00 | $4^{16}$ | $4^4$ | 8448 | 4.8 | 256 | 0.3 | 1 | 17 | 0.01 | 9 | 0.01 |
| | 0.95 | $4^{16}$ | $\dot{4}^9$ | 8448 | 4.8 | 2816 | 1.7 | 1 | 17 | 0.01 | 10 | 0.01 |
| | 0.93 | $4^{16}$ | $\dot{4}^{15}$ | 8448 | 4.8 | 7424 | 4.3 | 1 | 17 | 0.01 | 17 | 0.01 |
| | 0.90 | $4^{16}$ | $\dot{4}^{15}$ | $16\dot{9}$k | 102.2 | $14\dot{9}$k | 90.3 | 3 | 21 | 0.02 | 19 | 0.01 |
| | 0.85 | $4^{16}$ | $\dot{4}^{15}$ | $31\dot{3}$k | 184.4 | $27\dot{4}$k | 164.2 | 3 | 19 | 0.01 | 19 | 0.01 |
| CL | 1.00 | $4^{10}$ | $4^2$ | 368 | 5.0 | 16 | 0.5 | 1 | 36 | 0.5 | 5 | 0.1 |
| | 0.97 | $4^{10}$ | $\dot{4}^9$ | 684 | 7.9 | 620 | 7.1 | 1 | 36 | 0.5 | 32 | 0.4 |
| | 0.95 | $4^{10}$ | $\dot{4}^9$ | 1846 | 21.0 | 1677 | 18.2 | 3 | 33 | 0.5 | 30 | 0.4 |
| | 0.93 | $4^{10}$ | $\dot{4}^9$ | 2254 | 25.1 | 2048 | 22.8 | 2 | 30 | 0.4 | 27 | 0.4 |
| | 0.90 | $4^{10}$ | $\dot{4}^9$ | 2268 | 25.4 | 2060 | 25.0 | 2 | 30 | 0.5 | 27 | 0.4 |
| W | 1.00 | $4^{15}$ | $4^0$ | 61 | 0.9 | 1 | 0.1 | 1 | 25 | 0.4 | 1 | 0.1 |
| | 0.97 | $4^{15}$ | $\dot{4}^1$ | 61 | 0.9 | 3 | 0.1 | 1 | 25 | 0.4 | 1 | 0.1 |
| | 0.95 | $4^{15}$ | $\dot{4}^5$ | 61 | 0.9 | 13 | 0.3 | 1 | 25 | 0.4 | 5 | 0.1 |
| | 0.93 | $4^{15}$ | $\dot{4}^5$ | 1489 | 21.7 | 179 | 4.1 | 25 | 46 | 0.7 | 5 | 0.2 |
| | 0.90 | $4^{15}$ | $\dot{4}^5$ | $2\dot{4}$k | 359.1 | 729 | 42.1 | 197 | 46 | 0.7 | 5 | 0.2 |

Table 1: Comparison of different methods via the number of policies evaluated (#) and runtime (RT) in minutes. Numbers with a dot are approximate.

are similarly defined. $\mathcal{R}_R$ and $\mathcal{R}_R^H$ are shown in Figs. 4(a) and 4(b), respectively, for the larger domain. We created a small $4 \times 5$ domain (CS) for the exact methods and a large $4 \times 100$ domain (CL) for approximate solutions. To apply approximate solutions to CL, the states were aggregated based on features such as distance to the cliff, agent's position in the grid (e.g., along the edge or at the ends). For CL, we aggregated all non-terminal states into 10 clusters and retained the terminal states as is.

2) *Wumpus World (W)*: The agent is required to exit a $5 \times 5$ cave while collecting gold coins on its way out and avoiding encounters with the wumpus (i.e., staying in the same location) (see Fig. 3). The wumpus always chooses moves towards the agent. Collecting each gold coin gives a reward of $+30$. The game terminates if the agent encounters the wumpus ($-100$) or if it exits the cave ($+100$). In the ground-truth model ($\mathcal{M}_R$), the agent's actions are deterministic while the wumpus's actions are stochastic. The human's belief $\mathcal{M}_R^H$ is that the actions of both agents are stochastic. Under such a belief, the human would consider it dangerous for the agent to come close to the wumpus. For approximate solutions, the non-terminal states were aggregated into 15 clusters based on features such as the relative direction of the wumpus from agent and collection status of the gold coins.

**Results.** 1) *Performance Comparison*: Tab. 1 shows the runtime (except for BF and BF+ due to the large number of policies) and number of policies expanded (or evaluated) by each method under the three simulation domains. The first set of results is obtained with the small cliff world (CS) domain with 16 non-terminal states and 4 actions available in each state, resulting in $|\Pi| = 4^{16}$ policies. The second set of results is obtained with the large cliff world (CL) with 301 non-terminal states and 4 actions available in each state, resulting in $|\Pi| = 4^{301}$ policies. Applying state aggregation with 10 clusters results in $4^{10}$ policies. The third set of results is obtained with the wumpus world (W) with 2116

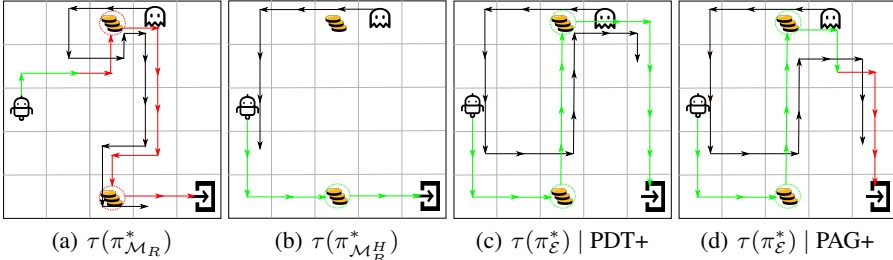

(a) $\tau(\pi^*_{\mathcal{M}_R})$   (b) $\tau(\pi^*_{\mathcal{M}^H_R})$   (c) $\tau(\pi^*_{\mathcal{E}})$ | PDT+   (d) $\tau(\pi^*_{\mathcal{E}})$ | PAG+

Figure 3: Behavior comparison in the wumpus world. Black lines show the trajectories of the wumpus. Red line segments show the parts of the agent's trajectories when the wumpus is in an adjacent cell, and green line segments show when the wumpus is at least two steps away. Presented are the most likely trajectories by (a) the optimal agent's policy, (b) the human's expectation, and the safe explicable policies obtained under $\delta = 0.90$ by (c) PDT+ and (d) PAG+, respectively.

non-terminal states and 4 actions available in every state, resulting in $|\Pi| = 4^{2116}$ policies. Applying state aggregation with 15 clusters results in $4^{15}$ policies. For CL and W, we use the approximate solutions for PDTs and PAGs.

We can observe that action pruning reduces the policy space and thereby the number of policies expanded by BF+, PDT+, and PAG+ as compared to those by BF, PDT, and PAG, respectively. The expansion order of policies in PDTs results in considerable additional pruning compared to BF+. With or without action pruning, PAGs expand fewer policies than PDTs since they only need to return a single policy. Lastly, while the number of policies expanded in PDTs increases (for lower $\delta$), it is interesting to note that PAGs sometimes expand fewer policies due to their greedy nature.

2) *Behavior Comparison in Cliff Worlds*: The results of the cliff worlds are shown in Figs. 5 (CS) and 4 (CL). Both the small and large domains introduce similar behaviors: shown only in the large domain, the optimal behavior in the agent's model takes the shortest path (Fig. 4(a)) whereas the human's expectation is to stay as far away from the cliff as possible (Fig. 4(b)). For SEP, Fig. 5 shows all the three policies in the Pareto set obtained given $\delta = 0.90$ in the small domain. Fig. 4(c) shows the most likely trajectories resulting from 2/3 policies in the Pareto set obtained given $\delta = 0.95$ in the large domain using the approximate solution. In general, we observe that the safe explicable policies result in trajectories that steer the agent away but not too far from the cliff to satisfy the bound while aligning with the human's expectation. In cliff worlds, to choose from $\Pi^*_{\mathcal{E}}$, we assign higher scores to policies producing simpler behaviors (e.g., fewer turns), it led to choosing the policy producing the green trajectory in Fig. 4(c) and the policy in Fig. 5(a). PAGs, on the other hand, computed different policies in $\Pi^*_{\mathcal{E}}$ (see figures).

3) *Behavior Comparison in Wumpus World*: The results are shown in Fig. 3. Under the optimal behavior in the agent's model ($\mathcal{M}_R$), the agent collected both coins while staying within the proximity of the wumpus before exiting, as shown in Fig. 3(a). The human's expectation (under $\mathcal{M}^H_R$) is that the agent avoids getting close to the wumpus and collecting a single coin before exiting, as shown in Fig. 3(b). When applying SEP under the bound $\delta = 0.90$, PDT+ returns a large Pareto set (see Tab. 1). To select from $\Pi^*_{\mathcal{E}}$, we score policies based on the average distance from the wumpus throughout the most likely trajectory. Fig. 3(c) shows the trajectory obtained from the policy with the highest score in

$\Pi^*_{\mathcal{E}}$: the agent managed to collect both coins while maintaining a cautious distance from the wumpus while taking a longer path, which is more explicable than the optimal agent's behavior in Fig. 3(a) and simultaneously more efficient than the human's expection in Fig. 3(b). Fig. 3(d) shows the behavior obtained by PAG+, which also maintains a cautious distance from the wumpus for the most part.

**Physical Robot Experiment**

**Robot Assistant Domain.** We implemented a scenario similar to the motivating example where a Kinova MOVO robot is assisting a human user with setting up the dining table (Fig. 6). The robot is required to fetch a napkin for the user from another table. The user does not fully understand the kinematic constraints of the robot arms and hence expects the robot to reach any location within its arm's length. Hence, the human's expectation is for the robot to place the napkin next to the plate (close to her). In the robot's model, however, movement of the arms is restricted by a vase on the table such that placing the napkin close to the user may tip over the vase containing water, resulting in a safety risk. Therefore, the robot's optimal behavior is to place the napkin next to the vase, which is further away from the user.

This experiment was run in a discretized setting where the state space was specified by the following variables: location of the robot, location of the napkin, and location of the vase. For any transition between the discrete states, the robot trajectories were generated a priori using *Move It*. More specifically, $\mathcal{M}^H_R$ specifies that the robot can access any location on the dining table irrespective of its location or the vase's position whereas $\mathcal{M}_R$ properly captures the influences from these factors. Our aim is to demonstrate that a robot running SEP would choose a costlier policy under $\mathcal{M}_R$ to be explicable to the human user while ensuring safety.

**Results.** The safe explicable behaviors from the results of the robot experiment are shown in Fig. 6. The optimal behavior under $\mathcal{M}_R$ involved two steps where the robot picked up the napkin and placed it on the table next to the vase and away from the user (see appendix B in supplemental materials). We then ran SEP with two different bounds that resulted in two different safe explicable behaviors. When the bound was $\delta = 0.85$ (Fig. 6(a)), the robot picked up the napkin, moved its entire body to be closer to the user so that it was no longer obstructed by the vase, before placing the napkin

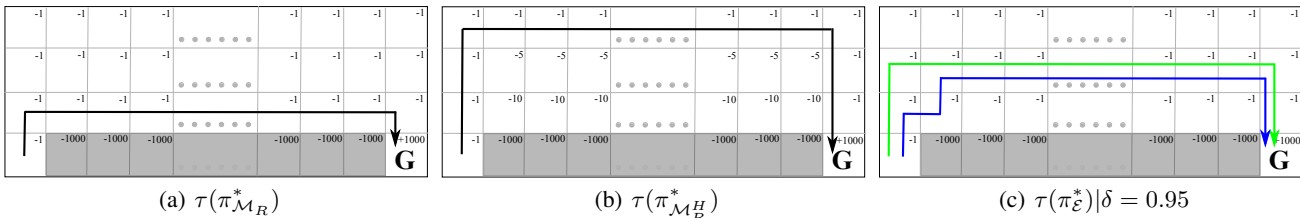

|  |  |  |
|---|---|---|
| (a) $\tau(\pi^*_{\mathcal{M}_R})$ | (b) $\tau(\pi^*_{\mathcal{M}_R^H})$ | (c) $\tau(\pi^*_{\mathcal{E}})|\delta = 0.95$ |

Figure 4: Behavior comparison in the large cliff world. Grey areas is the cliff and G is the goal. Reward for each state is shown at the top right corner. Displayed are the most likely trajectories from policies: (a) the optimal policy under $\mathcal{M}_R$, (b) the optimal policy under $\mathcal{M}_R^H$ (i.e., human expectation), (c) the safe explicable policies returned by PDT+ (green) and PAG+ (blue).

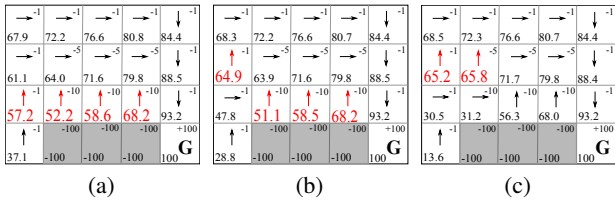

|  |  |  |
|---|---|---|
| (a) | (b) | (c) |

Figure 5: Pareto set obtained by PDT+ under $\delta = 0.90$ in the small cliff world. The state values are displayed under $\mathcal{M}_R^H$. Values highlighted in red are those that result in non-dominated policies. (b) shows the policy obtained by PAG+.

next to the plate. When the bound was set to $\delta = 0.80$ (Fig. 6(b)), the robot first pushed the vase aside so that it was no longer obstructing its arm movements before picking up the napkin and placing it next to the plate.

## DISCUSSION AND CONCLUSIONS

In this paper, we introduced the problem of Safe Explicable Planning (SEP), which significantly extends explicable planning to support a safety bound. To focus on the planning challenges of introducing safety in explicable planning, we assume the human belief model $\mathcal{M}_R^H$ is known. We provide references to existing literature where a similar assumption is made. When $\mathcal{M}_R^H$ is unknown, it can be learned by querying human subjects, to learn a reward function such as the approaches discussed in the survey paper (Wirth et al. 2017) and to learn domain dynamics such as the approach by (Zhuo 2015). We defer the consideration of other forms of $\mathcal{M}_R^H$ (such as hierarchical models) or learning $\mathcal{M}_R^H$ for future work. It is worth noting that the problem of SEP can be formulated as a CMDP problem under the special case where $\mathcal{T}_R = \mathcal{T}_R^H$ and $\gamma_R = \gamma_R^H$, in planning (Altman 2021) or learning (Achiam et al. 2017) setting.

In our work, we assume that safety is correlated to the expected return in the agent's model under the intuition that unsafe behaviors would result in low returns. Thus, imposing a lower bound on the return prevents unsafe behaviors. Such a safety definition is based on the constrained criterion. We aim to extend it to consider other safety formulations such as the worst-case criterion etc. outlined in (García and Fernández 2015) in the future. It would also be interesting to study how our value function-based criterion can be compared with or potentially equated to a state-machine-based criterion such as that studied by (Hunt et al. 2021). For example, prior work has studied how LTL constraints can be approximately considered by shaping the reward function (Camacho et al. 2019; Li, Vasile, and Belta 2017). Fur-

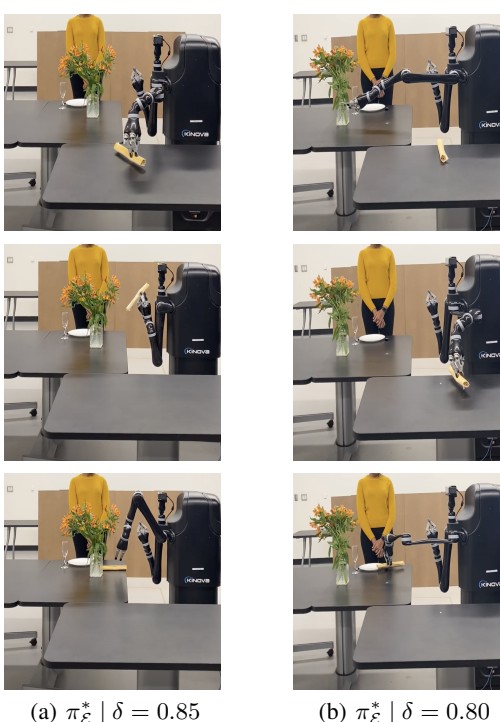

|  |  |
|---|---|
| (a) $\pi^*_{\mathcal{E}} \mid \delta = 0.85$ | (b) $\pi^*_{\mathcal{E}} \mid \delta = 0.80$ |

Figure 6: Safe explicable behaviors generated by PAG+ in the robot assistant domain under different bounds.

ther, the safety bound in our work is domain-specific as the criticality of safety is different in different scenarios and can be specified by the designer. We discuss how the safety bound was determined for the experimental evaluations and will more systematically address it in future work.

Our problem formulation generalizes the consideration of objectives from multiple models and the solution is a Pareto set of policies. We proposed an action pruning technique to reduce the search space, an exact method that returns the Pareto set, and a greedy method that returns a single policy. Existing literature in MOMDP shows that finding exact Pareto solutions for complex problems is intractable. To scale to complex domains, we further discussed approximate solutions by clustering states that are alike in terms of action selection. Since SEP requires searching the policy space, the methods proposed (exact and approximate) are still susceptible to policy explosion in large domains. The aggregation-based approximation proposed is preliminary work toward finding approximate safe explicable policies. We defer the study of generalized and more efficient approximation techniques to solve SEP in the future.

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
