# OpenReview forum: "Safe Explicable Planning"
_icaps-conference.org/ICAPS/2024/Conference — ICAPS 2024_

### Official Review · Reviewer_ZBQs · 2024-01-22

**Significance And Importance:** 2
**Soundness:** 3
**Novelty:** 3
**Clarity:** 4
**Overall Evaluation:** 2
**Confidence:** 2

**Weaknesses:**

2: No major or minor weaknesses.

**Contributions Of The Paper:**

The paper presents a concept of "Safe Explicable Planning" that, in a nutshell, concerns decision making of an agent that adheres to a safety bound (i.e., the agent has to act safely) and the behavior of the agent is understandable to a human. The problem is formulated as two MDPs, one for the agent and one for the human, such that both models share the same state and action space (but differs in the transition function, reward, and discount factor). To generate a set of all Pareto optimal policies, the paper presents a method called, Policy Descent Tree Search (PDT) and its variant enhanced by action pruning (PDT+). As it is argued that it might be impractical to enumerate all Pareto optimal policies, the paper introduces Policy Ascent Greedy Search (PAG) that computes only one Pareto optimal policy. The introduced methods are evaluated on two domains (Cliff Worlds and Wumpus World) and, on top of that, on the Robot Assistant Domain on a physical robot.

**Ethical Considerations:**

(1) Not Applicable: The paper does not have any ethical considerations to address

**Nomination For Best Paper:**

No

**Questions For Authors:**

1. Although explicit explainability is out of scope of the paper, I wonder how easy/hard is to make resulting policies (and decisions) explainable to humans ?
2. Can the methods be easily extended to consider multiple humans with different models ?

====
Thank you for your response !

**Reproducibility:**

3: Authors describe the implementation and domains in sufficient detail.

**Strengths Of The Paper:**

I like the idea of "Safe Explicable Planning" as it combines the importance of making safe decisions of the agent or the robot side while maintaining understandable and expectable behavior for humans. The core idea lies in considering two MDP models, one for the agent and one for a human, that share the state and action space. The PDT (and PAG that derives from PDT) starts from the optimal policy for the agent's model and then search through "worse" policies for those that maintains the safety bound and are not dominated by other in the human's model. Such methods are well designed, in my opinion, and provide theoretical guarantees (e.g. safety).

Experiments and well conducted and discussed. I especially like the experiment with physical robot (the Robot Assistant Domain) that illustrates the practical usefulness of the introduced methods.

The paper is generally well written and the contribution is well motivated.

**Weaknesses Of The Paper:**

Although it is elaborated how SEP differs from MOMDP and why the latter might not be ideal to tackle the problem, I feel that there might have been an empirical comparison with it (brute force policy search as a baseline might be a bit naive).

---

> ### Author Rebuttal · Authors · 2024-01-28
>
> Q1) By “explicit explainability”, we assume you mean explicitly communicating to explain a behavior. Explicit explainability would require generating explanations to the human user which is non-trivial. There exists prior work such as (Chakraborti et. al. 2017, Sreedharan et. al. 2017) that addresses this problem in deterministic environments. However, explaining the policies in stochastic environments such as (Finkelstein et. al. 2022, Topin et. al. 2019) is more complex as it involves several considerations such as attributing decisions to environment dynamics, generating selective explanations in the states visited by the agent (as explaining the entire policy may not be intuitive to the human), modeling the change in expectations of the human user as and when explanations are received by the human, etc.
>
> Q2) When learning the human model, multiple humans who have diverse preferences may be queried to learn a single human expected model (similar to (Xue et. al. 2023)) where SEP methods are still applicable. However, when there are multiple conflicting human models available, the scope of the problem would be required to extend to include a belief over these multiple models. Finding a safe explicable policy under such a belief over multiple human models is an interesting future direction to consider.
>
>
>
> Chakraborti, T., Sreedharan, S., Zhang, Y., and Kambhampati, S. Plan explanations as model reconciliation: moving beyond explanation as soliloquy. IJCAI 2017.
>
> Sreedharan, S., & Kambhampati, S. Balancing explicability and explanation in human-aware planning. AAAI Fall Symposium Series 2017.
>
> Finkelstein, M., Liu, L., Kolumbus, Y., Parkes, D. C., Rosenschein, J. S., & Keren, S. Explainable Reinforcement Learning via Model Transforms. NeurIPS 2022.
>
> Topin, N., & Veloso, M. Generation of policy-level explanations for reinforcement learning. AAAI 2019.
>
> Xue, W., An, B., Yan, S., & Xu, Z.. Reinforcement Learning from Diverse Human Preferences. arXiv preprint arXiv:2301.11774. 2023.

---

### Official Review · Reviewer_Hhck · 2024-01-23

**Significance And Importance:** 2
**Soundness:** 3
**Novelty:** 2
**Clarity:** 4
**Overall Evaluation:** 1
**Confidence:** 5

**Weaknesses:**

1: Minor weaknesses that are easily fixable.

**Contributions Of The Paper:**

The paper formulates the problem of Safe Explicable Planning (SEP) as finding a plan that maximizes the reward for the user's model of the agent (close to expectation) while adhering to some safety bound.  The safety bound is encoded as part of the reward function for the agent, so this constraint is manifest as a percentage of the optimal reward for the agent.  The paper first points out that  the space of possible policies can be reduced by removing all transitions that would violate the safety constraint.  The paper then develops two methods of searching this reduced policy space for the problem:  1) Policy Descent Tree Search (PDT), which starts from the agent's optimal policy (which satisfies the safety constraint) and explores all suboptimal actions from different states (as long as the safety constraint remains satisfied); and 2) Policy Ascent Greedy Search (PAG), which again starts with the optimal agent policy, but greedily explores one state/action change at a time that improves the user reward function.  Proofs sketches are provided that  PDT returns all Pareto optimal policies, and that PAG returns a policy in the Pareto optimal set.  In addition, state aggregation is employed for larger domains to collapse states that are "alike" according to some domain dependent feature.

PDT and PAG are evaluated  on three problems, small and large cliff worlds (CS & CL), and Wumpus World (W) for different safety bounds.  State aggregation was used for the two larger problems, CL and W.  An additional experiment was performed with a robot placing a napkin but avoiding the danger of tipping over a vase.

**Ethical Considerations:**

(1) Not Applicable: The paper does not have any ethical considerations to address

**Nomination For Best Paper:**

No

**Questions For Authors:**

1) Please comment on the difference in reward between PDT+ and PAG+ policies

2) It seems like there is a potential to guide the PDT and PAG search by focusing on those states where the user and agent optimal policies  are significantly different in action choice, but the user's action still satisfies the safety bound.  This can perhaps be seen as a way of measuring user "regret".  Please comment on the extent to which you already take advantage of this in your existing search.

3) Along the same lines, low regret in states could potentially be used as a basis for selection of states to aggregate.

4) In most cases, I don't think you care about the Pareto frontier, only in the optimal user policy still satisfying the safety constraint.  You can compute the optimal reward for the user, given the reduced policy space.  This would give you a bound on what could be attained.  If you find a policy "close enough" to this bound you could stop the search.  This would provide you with a search procedure that finds a bounded optimal policy subject to the safety constraints.  Seems worth considering as an intermediate between PDT and PAG.  Please comment.

**Reproducibility:**

4: Authors promise to release code and domains (whichever apply).

**Strengths Of The Paper:**

The paper clearly describes the problem, techniques, and experiments.  The example problems, although limited, illustrate interesting behavior, and illustrate that with state abstraction, the approach is feasible for small to medium sized domains.

**Weaknesses Of The Paper:**

1) The experimental results are limited to 3 fairly small synthetic problems and a small robot problem.  Two of the synthetic problems are identical in structure.

2) Your results do not show the difference in reward between the PDT(+) and PAG(+) policies.  This is important!

3) You never compare your techniques with/without different levels of state aggregation.

4) State aggregation appears essential for even moderate sized problems, and so far, this is a manual process.

5) In the Cliff problems, the agent model has a lower assessment of risk for walking next to the cliff than the user.  I was expecting the opposite (like in the robot problem).  I think you would get much more interesting behavior if there was varying risk at different places along the cliff, the users expectation was to walk along the cliff, and the SEP policy deviates away from the riskiest portions of the cliff depending on the safety bound.

6)  Related work: There is a close connection between the problem of SEP and OverSubscription Planning (OP).  In OP, different goals have different rewards, and there are resource limitations.  As a result, one is trying to optimize reward while remaining within the resource bounds.  The planning problem can be either stochastic or not.   There is also work on risk bounded planning done by Brian Williams and probably others.

---

> ### Author Rebuttal · Authors · 2024-01-28
>
> Q1) The different Q values of the policies under the human’s model are included in Fig. 5 (all Pareto policies from PDT+) and 5(b) (the greedy policy from PAG+), respectively, for CS domain. States in which the Q values differ are highlighted in red. As for the returns under the agent’s model, since we only focus on whether the safety bound is satisfied, we did not compare them explicitly but expect a similar relationship to hold as for the returns under the human’s model.
>
> Q2) In our current formulation, whether a policy is explicable or not only depends on how close the policy's return is to the optimal return in the human's model, assuming a rational observer. Given such an assumption, guiding the search by focusing on states where the actions differ (under some optimal policies in the human’s and agent’s models, respectively) may not be informative since two different policies may yield the same return. In particular, there may be different SEP policies that are similar in their returns but choose different actions under the same states. Hence, we did not consider differences between the optimal policies (under the two models) to guide the search in SEP.
>
> However, rationality is a strong assumption since humans are known to have preferences (including biases) in their expectations. In such cases, preference models must be learned and the “regret” suggested may indeed be consistent with these models under certain conditions.
>
> Q3) We plan to consider various methods for state aggregation in future work. It should be noted that the aggregation in SEP is meant to identify similar states for action selection: all states in a cluster are conditioned to choose the same action. Hence, it should generally be based on properties of the state itself rather than contrastive properties, such as “regret”, which is why we use state features in our evaluation. On the other hand, a contrastive property may inform the state property indirectly. Hence, it would be interesting to investigate in the future if considering “regret” along with state features may improve aggregation.
>
> Q4) Using a bound on explicability (return for optimal human’s policy) may potentially expedite the search, which is an interesting idea and can serve as an approximate solution when time is limited. However, finding such a bound and treating it as a soft constraint would be a challenge. An anytime search may also be used.
>
> W6) We will consider adding the connections suggested to related work.

---

### Official Review · Reviewer_CC5K · 2024-01-23

**Significance And Importance:** 2
**Soundness:** 3
**Novelty:** 3
**Clarity:** 4
**Overall Evaluation:** 2
**Confidence:** 4

**Weaknesses:**

0: Minor weaknesses requiring some work to be addressed for the paper to be accepted.

**Contributions Of The Paper:**

The primary contribution of the paper is the conceptualization and development of Safe Explicable Planning (SEP). This framework uniquely integrates safety constraints into the domain of explicable planning, a field that traditionally focuses on aligning robotic actions with human expectations. The SEP model goes a step further by ensuring that these actions also adhere to predefined safety norms, thereby enhancing the applicability to human-aware environments. Specifically, the approach involves formulating the problem as a Constrained Markov Decision Process and proposes a method to find a Pareto set of policies that balance explicability and safety. Additionally, the paper contributes methods for policy space reduction and state aggregation to enhance scalability. The paper also contributes computational simulations and physical robot experiments​​​​.

**Ethical Considerations:**

(1) Not Applicable: The paper does not have any ethical considerations to address

**Nomination For Best Paper:**

No

**Questions For Authors:**

1) How does the assumption of human rationality in your model affect its applicability and reliability in real-world scenarios, where human behavior can be unpredictable or non-rational?

2) Can you elaborate on the performance and scalability of the SEP methods in more complex environments? Are there limitations to the current state aggregation approach in these contexts?

3) Is there a trade-off between ensuring safety and maintaining high levels of explicability in the SEP model? How do safety constraints influence the agent's behavior from a human perspective?

**Reproducibility:**

5: Code and domains (whichever apply) are already publicly available

**Strengths Of The Paper:**

1. Integration of Safety in Explicable Planning:
Most current explicable planning models primarily focus on the alignment of robotic actions with human expectations, often overlooking the essential aspect of safety. By integrating safety, the paper addresses a significant gap, ensuring that AI behaviors are not only understandable to humans but also adhere to safety constraints, which is vital in real-world applications.

2. Soundness and Rigor of Methods:
The development of both exact and greedy methods for Safe Explicable Planning is methodologically sound. The exact method provides a comprehensive approach, while the greedy method offers a more computationally efficient alternative. The inclusion of formal proofs adds to the theoretical robustness of the paper.

3. Evaluation:
The authors have conducted both simulation-based evaluations and physical robot experiments. This dual approach demonstrates the practical applicability of the proposed methods in controlled environments and real-world scenarios, thus providing some empirical evidence to support the theoretical claims.

**Weaknesses Of The Paper:**

1. Assumption of Human Rationality and Other Human Factors:
While acknowledged as limitation, the paper assumes that humans interacting with the AI system will behave rationally. This assumption is a potential weakness because human behavior can be influenced by a range of factors and may not always be predictable, or even rational. In real-world settings, this could affect the performance and reliability of the proposed SEP model. Additionally, the paper does not consider other critical human factors, such as trust, that can impact the efficacy and applicability of SEP. For example, if the system's actions are not only explicable but also demonstrably safe, it may enhance user trust. Conversely, a lack of clear safety protocols could erode trust, even if the system's actions are explicable.

2. Scalability in Complex Environments:
While the paper addresses scalability through state aggregation, it may face challenges in more complex or dynamic environments where the state space can become significantly large. This raises concerns about the practical applicability of SEP in such scenarios and its ability to maintain computational efficiency and accuracy. As such, this question should be explored.

3. Evaluation in Limited Contexts and Generalizability:
Although the paper includes simulations and physical robot experiments, these are conducted in relatively controlled or limited contexts. Therefore, the results may not generalize. Different contexts or types of human-robot interactions could present unique challenges not accounted for in the current experiments, potentially affecting the broader applicability of the SEP approach.

---

> ### Author Rebuttal · Authors · 2024-01-28
>
> Q1 & W1) Under the assumption that humans are rational observers, we assume that a behavior/policy with a higher expected return in the human's model is more expected. Addressing irrationality and unpredictability requires learning the human's cognitive model for generating expectations, which is interesting but beyond the scope of this work.
>
> In our work, we focus only on a single interaction (human observing the agent’s behavior in a given task). Addressing the interaction between trust and SEP behavior involves longitudinal modeling, which is an interesting direction for future work.
>
> Q2 & W2) In SEP, when clustering is applied, the performance of the algorithms depends on the size of the aggregated state space. The approximation technique is susceptible to policy explosion if the resulting aggregated state space is still large as the method requires searching through the policy space, albeit reduced due to state space reduction. As mentioned in the discussion section of our paper, the proposed aggregation-based approximation is preliminary for safe explicable planning. The need for more efficient and generalized approximation methods is important for practical applications of SEP in very large MDPs, which we defer to future work.
>
> Q3) In SEP, maximizing explicability is subject to the safety constraint. It is likely that the most explicable behavior does not satisfy the safety constraint, which is precisely the reason we need SEP. In such a case, SEP may result in a behavior that is safe but less explicable to the human. Generally, because the human (user) may not be aware of the safety constraints encoded in the agent’s planning system, observing such behavior may lead to confusion. Explanations may be needed when the human questions the agent's behavior. For example, (Chakraborti et. al. 2017) considered generating an explanation for the agent’s plan. While explanation generation is also a key requirement of human-agent teaming, it is out of the scope of this paper.
>
> Chakraborti, T., Sreedharan, S., Zhang, Y., and Kambhampati, S. Plan explanations as model reconciliation: moving beyond explanation as soliloquy. IJCAI 2017.

---

### Meta-Review · Area_Chair_bKN3 · 2024-02-06

**Recommendation:** Accept (Poster)
**Confidence:** 4

**Metareview:**

This paper is recommended to be accepted; we encourage the authors to consider the comments made by reviewers to ensure the best possible camera-ready version is submitted.

**Ethical Considerations:**

(1) Not Applicable: The paper does not have any ethical considerations to address